# Combined Jasmonic Acid and Ethylene Treatment Induces Resistance Effect in Faba Bean Plants Against *Frankliniella occidentalis* (Pergande) (Thysanoptera: Thripidae)

**DOI:** 10.3390/insects13111073

**Published:** 2022-11-21

**Authors:** Yu-Lian Jia, Tao Zhang, Jun-Rui Zhi, Lu Tuo, Wen-Bo Yue, Ding-Yin Li, Li Liu

**Affiliations:** 1Institute of Entomology, Guizhou University, Guiyang 550025, China; 2Laboratory for Agricultural Pest Management in the Mountainous Region, Guiyang 550025, China

**Keywords:** *Frankliniella occidentalis*, faba bean (*Vicia faba* L.) plants, inducible plant defense, exogenous phytohormones, life table

## Abstract

**Simple Summary:**

The western flower thrips *Frankliniella occidentalis* (Thysanoptera: Thripidae) is a major pest of vegetable and flower crops worldwide. A feasible and environmentally friendly method to control *F. occidentalis* is to induce resistance in host plants. In this study, we attempted to induce resistance against *F. occidentalis* in faba bean plants. We first investigated the effect of different concentrations of jasmonic acid (JA) and ethylene (ET) alone on the oviposition of *F. occidentalis*. We then used different concentrations of the combined treatment (JA and ET) to evaluate the most suitable concentrations affecting the oviposition and feeding selectivity of *F. occidentalis*. Second, we examined the effects of mixed exogenous JA and ET treatments on the activities of JA and ET biosynthetic enzymes in faba bean leaves and on the expression levels of JA/ET pathway marker genes. Finally, we evaluated the effects of JA/ET treatment-related induced resistance on the life history and population parameters of *F. occidentalis*. Our results provide reference data for the control of thrips infestation in faba bean crops via application of exogenous phytohormones.

**Abstract:**

*Frankliniella occidentalis* (Pergande) (Thysanoptera: Thripidae) is a serious invasive pest in China. In this study, we determined whether exogenous jasmonic acid (JA) and ethylene (ET) treatments could induce resistance against *F. occidentalis* in faba bean plants. First, we investigated the effects of different concentrations of JA or ET alone on *F. occidentalis* and then assessed the effects of optimal concentrations of JA and ET combined. Our results showed that the optimal concertation of JA was 2 mmol/L and ET was 0.5 mmol/L. JA + ET mixture showed the greatest inhibitory effect in terms of oviposition and feeding. JA with ET was found to induce changes in the activities of lipoxygenase (LOX), allene oxide synthase (AOS), polyphenol oxidase (PPO), 1-aminocyclopropane 1-carboxylic acid synthase (ACS), and trypsin inhibitor (TI). This treatment also activated or inhibited the relative expression levels of *LOX1*, *ACO2*, *ACS2*, and *AP2/ERF*. Treatment of faba bean plants with JA and ET significantly prolonged *F. occidentalis* development and adult preoviposition period, significantly reduced per-female oviposition, and altered male longevity and offspring demographic parameters. These results indicate that JA with ET can induce defenses against the growth and development of *F. occidentalis* in faba bean plants.

## 1. Introduction

Herbivory and associated injuries can induce plant defense functions via altered gene expression to trigger the production of secondary metabolites, plant defense proteins, and volatile compounds. Herbivory can also alter enzymatic activity [1,2]. Plant defenses can interfere with herbivore feeding, growth and development, fecundity, and fertility [3]. Studies have shown that the exogenous application of phytohormones triggers an effect similar to insect feeding [4].

Jasmonic acid (JA) and ethylene (ET) are two important phytohormones [5]. The exogenous application of phytohormones can lead to the accumulation of endogenous phytohormones and improve plant defense [6,7,8]. Exogenous JA treatments were reported to induce the accumulation of endogenous JA in asparagus [9]. Moreover, exogenous JA treatments induced the release of ethylene in tomatoes [10]. The JA biosynthetic pathway involves converting polyunsaturated fatty acids to *cis*- (+)-12-oxophytodienoic acid (OPDA) and dnOPDA via lipoxygenase (LOX), allene oxide synthase (AOS), and allene oxide cyclase (AOC). JA is then formed by oxophytodienoate reductase 3 (OPR3) followed by three cycles of b-oxidation [11]. ET biosynthesis progresses from S-adenosyl-L-methionine (SAM) via two specific enzymatic reactions, catalyzed by 1-aminocyclopropane-1-carboxylic acid (ACC), 1-aminocyclopropane 1-carboxylic acid synthase (ACS), and ACC oxidase (ACO) [12]. ET and JA can regulate different defense response processes independently, but they have also been reported to interact with each other [13]. 

The western flower thrips *Frankliniella occidentalis* (Thysanoptera: Thripidae) is one of the most destructive herbivorous insects worldwide. It can damage many types of crops, thereby causing huge economic losses [14,15]. The biological advantages of thrips include their small size, short developmental duration, and diverse assortment of potential host plants [16]. It is difficult to control *F. occidentalis* with traditional pesticide treatments because this species is resistant to many pesticides. In addition, the use of chemical agents can cause environmental pollution and pose a risk to human health [17]. JA plays a very important role in plant insect interaction defense against insects [18]. Not only exogenous spraying of JA, but inhibiting the JA metabolic pathway also causes changes in pest behavior [19]. Therefore, it is necessary to seek environmentally friendly control strategies capable of effectively reducing crop yield losses through this approach. One such method for the control of *F. occidentalis* involves inducing resistance in host plants. 

Studies on how exogenous phytohormone administration induces host plant resistance against *F. occidentalis* have mainly focused on model plants, with few studies on non-model species. For example, JA treatment enhanced resistance against thrips and limited thrips population development in sweet pepper [20], and MeJA treatment prolonged the survival time of tomato plants when exposed to thrips larvae [21]. In Arabidopsis, JA application enhanced tolerance against thrips and reduced thrips feeding [22]. JA treatment on kidney beans increased the activity of defense enzymes [23], prolonged the immature development period of thrips, and reduced the oviposition of thrips [24]. 

Faba bean is a major cash crop and a major winter host of thrips [25,26]. However, mechanisms underlying JA- and ET-induced resistance in faba bean plants remain unclear. To the best of our knowledge, this study is the first to assess the effect of different concentrations of JA or ET on the oviposition of *F. occidentalis* and to determine the optimal concentrations of exogenous treatment of each phytohormone. We also assessed the effects of the combination of the optimal concentrations of JA and ET on *F. occidentalis*. After determining the optimal combination of JA and ET treatment, with respect to its effect on the oviposition of *F. occidentalis*, we sprayed this combined treatment on faba bean plants, to verify whether it affected defense enzyme activity and the expression of JA/ET pathway marker genes. We also examined the effect of administration of these phytohormones on the two-sex life parameters of thrips. The results of this study provides useful empirical data for future studies on the mechanism by which JA and ET induce defense against *F. occidentalis* in faba bean plants.

## 2. Materials and Methods

### 2.1. Insect Rearing and Host Plant Cultivation

*F. occidentalis* specimens were collected from vegetable crops in Huaxi District, Guiyang City, China. After identification, insects were reared in a laboratory-grade multistage programmable artificial climate box (purchased from Ningbo Jiangnan Instrument Factory, Ningbo, China). *Phaseolus vulgaris* bean pods, purchased from a vegetable market (Xujiachong Farmers’ Market, Huaxi District, Guiyang City, China), were used to establish a population for the experiment. Thrips were reared in the laboratory for more than 50 generations.

Faba bean (*Vicia faba* L.), variety Lincan No. 5 (Jinzhong Agricultural Products Industrial Development Co., Ltd., Gansu, China), was used for all experiments. Faba bean seeds were planted in an artificial climate chamber in a sterilized nutrient soil nutrition pot; one seed was planted per pot. Cultivation parameters were as follows: temperature 25 °C ± 5 °C, relative humidity 75% ± 5%, and photoperiod 14 h/10 h (L/D). When plants reached 8–10 leaves, faba bean plants that showed consistent growth were selected as test plants for future experiments. No pesticides were used during the growth period and no pest damage occurred.

### 2.2. Different Concentrations of JA and ET Solutions 

JA was purchased from Sigma–Aldrich, Co. Ltd. (Shanghai, China). ET was prepared with ethephon purchased from Solarbio Biotechnology Co. Ltd. (Beijing, China). Before use, 105 µL of JA and 0.0722 g of ethephon were separately dissolved in 2 mL aliquots of ethanol, respectively. Each solution was then diluted with distilled water to a working concentration of 50 mL at 10 mmol/L. JA was further diluted with distilled water to concentrations of 0.1, 0.5, 1.0, 1.5, 2.0, and 2.5 mmol/L. Ethephon was further diluted with distilled water to concentrations of 0.1, 0.5, 1.0, 1.5, and 2.0 mmol/L.

We then screened the abovementioned five concentrations of both the phytohormones with respect to their effect on thrips oviposition. The most effective concentration that inhibited oviposition was chosen as the optimal concentration. We then created the following three different combined treatments using the optimal concentration of each phytohormone: (1) JA + ET: spraying JA first and then ET; (2) ET + JA: spraying ET first and then JA; (3) JA with ET: a mixture of JA and ET was sprayed at the same time. In the first two treatments, the interval between the two spraying applications was 48 h. Among these three treatments, the treatment that showed the greatest effect on the oviposition of thrips was chosen for following experiments.

### 2.3. Plant Treatments

Approximately 3.5 mL of different phytohormone solutions were evenly sprayed on each faba bean plant. Distilled water was used as a negative control. Sprayed faba bean plants were then covered with a transparent circular plastic bucket shroud with a height of 29.7 cm and a diameter of 11 cm. Further experiments were performed after 48 h from the last spray application. All experiments were performed in an artificial climate chamber programmed with the conditions noted above. 

### 2.4. Oviposition Assays

We used an experimental apparatus consisting of a disposable Petri dish with a diameter of 9 cm to support two faba bean leaves at the same position on different treatment plants. Ten female adult thrips (5-day-old adults) were released on the leaves of treated faba bean plants. Petri dishes were then immediately sealed with parafilm to prevent thrips from escaping. All adult thrips were removed after 24 h, and leaves with eggs (thrips deposit eggs in leaf tissue) were checked every day. The number of hatched larvae was used to measure the oviposition [27]. The total number of hatched larvae was determined after 6 days, to ensure that all eggs hatched. In this experiment, each treatment had 15 biological replicates.

### 2.5. Leaf Disc Thrip Selectivity Assays

Two days (48 h) after spraying, we obtained leaf discs from faba bean plants treated with JA + ET, ET + JA, JA with ET, or distilled water. Plastic boxes with a diameter of 18.5 cm and a height of 2.3 cm were divided into four equal parts for leaf disc selection. Four leaf samples were taken from each of the different treatments; circles were cut from petioles, wrapped in wet cotton, and then placed in the corresponding area of the plastic box. Larvae of 20 thrips (3-day-old larvae) were starved for 4 h and placed in the center of the leaf disc, equidistant from the leaves [20]. We counted the number of thrips on the leaves of faba bean plants at intervals of 6, 12, 24, and 48 h after the start of the trial. One leaf disk was considered one replicate, and we used a total of 33 replicates.

To further determine the effect of the JA with ET solution on thrips, we conducted another experiment assessing the selectivity of thrips between healthy untreated leaves and JA with ET treated leaves. For this test, leaf discs were divided into two equal parts. The leaf and thrips treatment procedures were the same as the disc method reported above, with a total of 25 replicates being used for each treatment.

### 2.6. Enzyme Activity Assays 

For enzyme activity assays, faba bean plants were treated with JA and ET for 48 h. Leaf samples (0.1 g) were then obtained and placed in 2.0-mL centrifuge tubes. After quenching with liquid nitrogen, the samples were stored at −80 °C for later use. The enzymatic activities of LOX and PPO in treated and untreated faba bean plants were determined using a kit from Suzhou Comin Biotechnology Co. Ltd. (Suzhou, China) following the manufacturer’s instructions [23]. ACS, AOS, and trypsin inhibitor (TI) activities were determined using a kit from Yaji Biotechnology Co., Ltd. (Shanghai, China) via an enzyme-linked immunosorbent assay. All procedures were performed as per the manufacturer’s instructions. The enzyme activities were calibrated to the protein content of the corresponding sample concentration. There were 4 replicates for each enzyme activity.

### 2.7. Expression Levels of JA/ET Pathway Marker Genes

We obtained samples from faba bean plants as above, and total RNA extraction was performed according to the instructions of the Eastep Super total RNA extraction kit (Promega Biotech Co., Ltd., Beijing, China). First strand cDNA was synthesized using the Thermo Fisher Scientfic RevertAid kit (Thermo Scientific, Vilnius, Lithuania). We used the CFX96TM real-time fluorescent quantitative PCR system (LongGene Scientific Instruments Co., Ltd., Hangzhou, China) and FastStart Essential DNA Green Master kit (Roche Diagnostics, Penzberg, Germany) to quantify gene expression. The total volume of the reaction mixture was 10 µL and it included 5.0 µL of FastStart Essential DNA Green Master (Roche Diagnostics GmbH, Penzberg, Germany), 0.5 µL of forward and reverse primers, 1 µL of cDNA, and 3 µL of dd water. The reaction procedure was as follows: predenaturation at 95 °C for 10 min, denaturation at 95 °C for 30 s, annealing at the specified temperature for 30 s (see annealing temperature primer list), 40 cycles, and final extension at 65 °C for 5 s to generate a melting curve and collect Ct values. Sequences were searched using the National Center of Biotechnology Information (NCBI) database. Primer version 6.0 (PREMIER Biosoft, Palo, CA, USA) was used for sequence alignment. Faba bean *ELF1A* was used as an internal reference gene [28], and we used designed primers for the related genes *LOX1* (GenBank accession no. Z73498), *ACO2* (GenBank accession no. EU543654) and *ACS2* (GenBank accession no. EU543656). Primers for *AP2/ERF* (GenBank accession no. XM_003611639) were obtained from published sequences [29]. All primer sequences are shown in Table 1 and were synthesized by Sangon Biotech Co. Ltd. (Shanghai, China). There were three replicates for each gene.

### 2.8. Life Table Study

A total of 20 thrips (5-day-old adults) were released on faba bean plant samples treated with both JA and ET for 48 h. Five faba bean plants were tested. After 10 h, the thrips were removed and the plants were checked twice a day (i.e., at 8:00 am and 8:00 pm) to assess the egg duration. Once the larvae hatched, they were immediately transferred to a new small plastic box for single feeding (d = 2.9 cm). Wet water filter paper was then added under the faba bean leaf sample (1 cm^2^). Fresh faba bean leaves were replaced every 1–2 days. Thrips development and survival rates were observed and recorded every 12 h [30]. Once adult thrips reached eclosion, a male and female were paired immediately and observed once a day, both female and male longevity was recorded until the thrips died. At the same time, new leaves were replaced every day, and the leaf with thrips eggs was transferred to a new box respectively. Leaves were continuously observed until no larvae hatched, and the number of hatched larvae was deemed to be the total number of eggs oviposited by thrips [27]. Larvae were continually fed with fresh faba bean leaves until male and female identification was possible; offspring sex ratios were then calculated. In the experiment, 20 thrips were included as a single replicate, and four replicates were produced for each condition. Thrips were collected and reared for later use. 

### 2.9. Data Analysis

Microsoft Excel 2019 was used to perform basic statistical analysis. Relative gene expression was calculated using the 2-^ΔΔCt^ quantitative method [31]. SPSS version 22.0 (IBM SPSS, Chicago, IL, USA) was used to perform one-way analysis of variance, and Tukey’s multiple comparison tests (α = 0.05) were used to assess the statistical significance of differences among treatments. Response variables of interest included oviposition, thrips selectivity, enzyme activity, and enzyme gene expression level. Origin 2018 (OriginLab Corporation, Northampton, Massachusetts, USA) and SigmaPlot version 14.0 (Systat Software, Inc., San Jose, CA, USA) were used to plot graphs.

Life table statistics were analyzed using Microsoft Excel 2019, and the associated graphs were plotted using SigmaPlot v14.0. For the age-stage, two-sex life table analysis [32,33], we used the computer program TWOSEX-MSChart to process the developmental data of *F. occidentalis* (0.5 d) (http://140.120.197.173/Ecology/) [34]. This program was used to calculate the age-stage-specific survival rate (*Sxj*, where *x* = age and *j* = stage), age-stage-specific life expectancy (*exj*), age-stage-specific reproductive value (*Vxj*), age-specific survival rate (*lx*), and age-specific survival rate [35,36]. Population parameters, including net reproductive rate (*R*_0_), mean generation time (*T*), intrinsic rate of increase (*r*), finite rate of increase (*λ*), gross reproductive rate (*GRR*), were calculated and their mean and standard error were determined using 100,000 bootstrap samples [37].

## 3. Results

### 3.1. Effects of JA and ET Treatments on the Oviposition of F. occidentalis

All concentrations of JA solution applied to faba bean plants showed an inhibitory effect on the oviposition of thrips (Figure 1A). The oviposition rates with JA treatment at concentrations of 0.1, 0.5, 1, 1.5, 2, and 2.5 mmol/L were 77.11%, 73.59%, 81.40%, 81.97%, 60.37%, and 73.88% of the control, respectively (*F*_6,98_ = 8.027, *p* < 0.01). The concentration of 2 mmol/L showed the greatest inhibitory effect.

Faba bean plants treated with ethylene also showed a significant inhibitory effect on the oviposition of thrips (Figure 1B) (*F*_5,84_ = 6.37, *p* < 0.01). The oviposition rates with ET treatment at concentrations of 0.1, 0.5, 1, 1.5, and 2 mmol/L were 73.32%, 67.93%, 82.48%, 83.57%, and 86.53% of the control, respectively. The greatest oviposition-inhibiting effect was observed at a concentration of 0.5 mmol/L.

### 3.2. Effects of Combined JA and ET Treatments on the Oviposition of F. occidentalis

Thrips oviposition on faba bean plants treated with different combinations of JA and ET was also significantly lower than that on negative control (Figure 2) (*F*_3,76_ = 36.27, *p* < 0.01). The three treatments, i.e., JA + ET, ET + JA, and JA with ET, showed thrips oviposition rates of 55.88%, 81.68%, and 50.65% of the control, respectively. The inhibitory effect was the strongest for the JA + ET and JA with ET treatments. Since the JA with ET treatment showed the greatest oviposition reduction, this treatment was used for further experiments.

### 3.3. Effect of Combined JA and ET Treatment on the Oviposition Selectivity of F. occidentalis on Faba Bean 

Three different combinations of JA and ET were found to affect the feeding selectivity of thrips (Figure 3A). For simultaneous treatment, the feeding selectivity of thrips was significantly lower than that of the control at 6, 12, and 24 h (*F*_3,128_ = 52.06, *p* < 0.01; *F*_3,128_ = 27.56, *p* < 0.01; *F*_3,128_ = 10.36, *p* < 0.01), and we observed no significant difference among treatments at 48 h (*F*_3,128_ = 2.032, *p* > 0.05). Other treatments (i.e., those in which phytohormones were applied at different times) showed different thrips selectivity patterns. The number of thrips in the control decreased gradually from 6 to 48 h (*F*_3,128_ = 9.52, *p* < 0.01), with the number being the lowest at 48 h. However, no significant difference in the number of thrips among the four time periods for both the JA + ET and ET + JA treatments was observed (*F*_3,128_ = 0.99, *p* > 0.05; *F*_3,128_ = 0.07, *p* > 0.05). For the JA with ET treatment, we observed the lowest number of thrips feeding at 6 h after treatment, and this number gradually increased over time, reaching its highest value after 48 h (*F*_3,128_ = 3.07, *p* < 0.05).

In the experiments that just focused on the effect of JA with ET, the JA with ET treatment was found to significantly affect thrips feeding selectivity, compared with the control (Figure 3B). The number of feeding larvae on faba bean plants treated with JA with ET at 6, 12, 24, and 48 h was significantly lower than in the control (*t* = 3.83, df = 48, *p* < 0.01; *t* = 3.03, *df* = 48, *p* < 0.01; *t* = 3.48, *df* = 48, *p* < 0.01; *t* = 2.06, *df* = 48, *p* < 0.05). For the treatments where the phytohormones were applied at different times, we found no significant changes in the number of thrips feeding (*F*_3,96_ = 0.167, *p* > 0.05; *F*_3,96_ = 0.184, *p* > 0.05).

### 3.4. Effect of the JA with ET Treatment on the Activities of Defense Enzymes in Faba Bean Plants

The LOX activity in faba bean plants treated with JA with ET was significantly higher than that in the control at 24 h and 72 h (Figure 4A) (*t* = 4.31, *df* = 6, *p* < 0.01; 37, *df* = 6, *p* < 0.05). However, this difference was not significant at 48 h (*t* = 1.55, *df* = 6, *p* > 0.05). Within the same treatment, observed activities changed for treatment time points. In the control, the LOX activity was significantly higher at 48 h than at 24 h and 72 h (*F*_2,9_ = 7.76, *p* = 0.01). For the JA with ET treatment, we found no significant difference in LOX activity among the three treatment time points (*F*_2,9_ = 2.43, *p* > 0.05).

The AOS activity in the plants treated with both JA and ET was significantly lower than that in the control at 24 h (Figure 4B) (*t* = 13.00, *df* = 6, *p* < 0.01). However, the AOS activity in the plants treated with both JA and ET was significantly higher than that in the control at 48 h and 72 h (*t* = 12.33, *df* = 6, *p* < 0.01; *t* = 20.24, *df* = 6, *p* < 0.01). Moreover, in the control, the AOS activity was significantly higher at 24 h than at 48 h and 72 h (*F*_2,9_ = 96.09, *p* < 0.01). However, after treatment with JA and ET, the AOS activity was significantly higher at 72 h than at 24 h and 48 h (*F*_2,9_ = 542.16, *p* < 0.01).

The PPO activity in plants treated with JA and ET increased significantly at 24 h and 48 h (Figure 4C) (*t* = 4.85, *df* = 6, *p* < 0.01; *t* = 6.24, *df* = 6, *p* < 0.01); however, no significant change was observed in the control after 72 h (*t* = 0.019, *df* = 6, *p* > 0.05). The PPO activity in the control did not change significantly over the three time periods (*F*_2,9_ = 0.20, *p* > 0.05), whereas the PPO activity in the plants treated with JA and ET was significantly higher at 48 h than at 24 h and 72 h (*F*_2,9_ = 27.69, *p* < 0.01).

The ACS activity in plants treated with JA and ET was significantly lower than that in the control at 24 h (Figure 4D) (*t* = 7.47, *df* = 6, *p* < 0.01). Moreover, no significant difference in the ACS activity was observed at 48 h (*t* = 0.82, *df* = 6, *p* > 0.05). However, the ACS activity was significantly higher than that in the control at 72 h (*t* = 6.12, *df* = 6, *p* < 0.01). In the control, the ACS activity at 24 h and 72 h was significantly higher than that at 48 h (*F*_2,9_ = 56.96, *p* < 0.01). In contrast, the ACS activity in the plants treated with JA and ET at 72 h was significantly higher than at 24 h and 48 h (*F*_2,9_ = 110.05, *p* < 0.01).

No significant change in the TI activity between JA- with ET-treated and control plants was observed (Figure 4E) at 24 h and 48 h (*t* = 1.87, *df* = 6, *p* > 0.05). The TI activity in plants treated with JA with ET significantly increased at 72 h (*t* = 6.49, *df* = 6, *p* < 0.01), and in both control and JA with ET-treated plants, we found that the TI activity was significantly higher at 72 h than at 24 h and 48 h (*F*_2,9_ = 23.52, *p* < 0.01, *F*_2,9_ = 248.94, *p* < 0.01).

### 3.5. Effect of the JA with ET Treatment on the Expression of JA/ET Pathway Marker Genes 

The relative expression of *LOX1* in the samples treated with JA and ET was significantly higher than that in the control at 24 h (*t* = 8.53, *df* = 4, *p* < 0.01) and significantly lower than that in the control at 48 h (*t* = 5.60, *df* = 4, *p* < 0.01). However, no difference in the relative expression of *LOX1* was observed at 72 h (*t* = 2.45, *df* = 4, *p* > 0.05). In the control, there was no significant difference in the relative expression of *LOX1* at 24 h, 48 h, and 72 h (*F*_2,6_ = 3.89, *p* > 0.05). However, we saw increased expression in plants treated with JA with ET at 24 h relative to 48 h, but this was not significantly higher than the level recorded at 72 h (*F*_2,6_ = 6.84, *p* < 0.05) (Figure 5A).

The relative expression of *ACO2* in plants treated with JA and ET was not significantly different at 24 h and 72 h compared with the control (*t* = 1.74, *df* = 4, *p* > 0.05; *t* = 1.17, *df* = 4, *p* > 0.05). However, we did observe a significant increase in the relative expression of *ACO2* at 48 h (*t* = 12.73, *df* = 4, *p* < 0.01). The relative expression of *ACO2* in the control was significantly higher at 24 h than at 48 h and 72 h, and there was no significant difference in the relative expression of *ACO2* at 48 h and 72 h (*F*_2,6_ = 28.72, *p* < 0.01). The relative expression of *ACO2* in plants treated with JA and ET at 48 h was significantly higher than at 24 h and 72 h. Moreover, we found that the relative expression of *ACO2* at 24 h was significantly higher than that at 72 h (*F*_2,6_ = 76.70, *p* < 0.01) (Figure 5B).

The relative expression of *ACS2* in plants treated with JA and ET was significantly lower at 24 h and 48 h than that in the control (*t* = 5.87, *df* = 4, *p* < 0.01; *t* = 20.85, *df* = 4, *p* < 0.01). The relative expression of *ACS2* was found to significantly increase at 72 h (*t* = 13.45, *df* = 4, *p* < 0.01). In the control, the *ACS2* activity was significantly higher at 48 h than at 24 h and 72 h, and the activity at 24 h was significantly higher than at 72 h (*F*_2,6_ = 632.78, *p* < 0.01). For the plants treated with JA and ET, the *ACS2* activity at 72 h was significantly higher than at 48 h and 24 h, and the *ACS2* activity at 48 h was significantly higher than that at 24 h (*F*_2,6_ = 154.28, *p* < 0.01) (Figure 5C).

The relative expression of *AP2/ERF* in plants treated with JA and ET showed no significant changes compared with the control at 24 h (*t*=1.09, *df* = 4, *p* > 0.05). However, the plants treated with JA and ET showed significantly higher *AP2/ERF* expression than the control at 48 h and 72 h (*t* = 64.88, *df* = 4, *p* < 0.01; *t* = 14.19, *df* = 4, *p* < 0.01). The relative expression of *AP2/ERF* in the control at 24 h was significantly higher than that at 48 h and 72 h, and we observed no significant change in the expression between 48 h and 72 h (*F*_2,6_ = 223.46, *p* < 0.01). For samples treated with JA and ET, we observed that the relative *AP2/ERF* expression at 48 h was significantly higher than that at 24 h and 72 h. We also found that the relative *AP2/ERF* expression at 24 h was significantly higher than that at 72 h (*F*_2,6_ = 183.09, *p* < 0.01) (Figure 5D).

### 3.6. Effect of the JA with ET Treatment on the Life Parameters of F. occidentalis

Treatment with JA and ET had a significant effect on thrips developmental duration and reproduction (Table 2). The egg (3.04 d), first instar (2.23 d), immature (14.91 d), preoviposition period (3.37 d) and total preoviposition period (18.56 d) stages of thrips were all significantly prolonged (*p* < 0.05). The oviposition per female (10.08 d) and male adult stage (9.29 d) was found to be significantly decreased (*p* < 0.05). We found no significant differences in the durations of the second instar stage (4.42 d), prepupal stage (1.46 d), pupal stage (3.83 d), and female adult stage (15.08 d). We also found no differences in female longevity (30.27 d), male longevity (23.95 d), and offspring sex ratio (1.00) compared with the control (*p* > 0.05).

### 3.7. Effect of the JA with ET Treatment of Faba Bean Plants on the Survival Rate and Fecundity of F. occidentalis 

The age-stage survival curve (*S_xj_*) represents the probability that a newly laid egg will survive to age *x* and stage *j* (Figure 6). The *S_xj_* overlap degree in faba bean thrips fed on the leaves of plants treated with JA and ET was lower than that in the control. We also found that the survival time of faba bean thrips fed on the leaves of plants treated with JA and ET was not significantly different from the control. The survival rate of first instar larvae was higher in treated plants than that in the control, but the survival rate of female and male adults was lower than that in the control. At the pupal stage, we found that the survival rate of male and female adults feeding on treated plant samples was lower than in the control; however, there was no significant difference between treatment and control plants at the egg, second instar, and prepupal stages.

Next, we examined the results of the age-specific survival rate (*l_x_*) curve (Figure 7). The (*l_x_*) curve of the JA with ET treatment began to show a downward trend after 7 days, whereas the control showed a downward trend after 20 days. The age-stage-specific fecundity (*f_x_*) of females peaked between 20 and 25 days, while that in females on the JA with ET-treated plants peaked between 20 and 30 days, with multiple peaks in the middle. The age-specific net maternity (*l_x_m_x_*) of the JA with ET treatment was significantly lower than that of the control.

The age-stage-specific life expectancy (*e_xj_*) curves are shown in Figure 8. The (*e_xj_*) curve life expectancy of thrips decreased with age. The age-stage-specific life expectancies of thrips in plants treated with JA and ET was lower than that in the control at all stages.

The age-stage-specific reproduction value (*v_xj_*) curve refers to the average contribution of thrips individuals at *x* age and *j* stage to the future population development (Figure 9). The reproductive value of thrips from the JA and ET and control groups showed first an increase and then a decrease with age. However, the reproductive peak in the treatment group was lower than in the control.

### 3.8. Effect of the JA with ET Treatment in Faba Bean Plants on the Life Parameters of F. occidentalis

Treatment with JA and ET resulted in significant changes in the life parameters of thrips (Table 3). The intrinsic rate of increase (*r* = 0.05), finite rate of increase (*λ* = 1.05), net reproductive rate (*R*_0_ = 3.3), and mean generation time (*T* = 23.77) of thrips feeding on JA and ET-treated plants were all significantly lower than those in the control (*r* = 0.08, *λ*= 1.08, *R*_0_ = 5.73, and *T* = 22.01). The total gross reproductive rate (*GRR* = 9.2) of treated plants was also lower than that of the control (*GRR* = 10.76), but this difference was not significant.

## 4. Discussion

Exogenous phytohormones exert a concentrated effect on plant defenses against herbivorous insects [38]. As important phytohormones, JA and ET play a role in regulating plant adaptations to biotic and abiotic stresses [39]. There was no report about any negative impact of exogenous JA and ET on plant quality. In this study, exogenous JA and ET did not adversely affect the growth and development of faba bean plants. The cost of using exogenous JA and ET is not higher than that of using traditional thrips management strategies. Moreover, insect pests have developed resistance. In order to improve the control effect, massive amounts of chemical pesticides have to be sprayed, which is more costly and causes pollution to the environment. In this study, we found that the oviposition of *F. occidentalis* was significantly reduced after different concentrations of JA and ET treatment, with 2.0 mmol/L JA and 0.5 mmol/L ET showing the greatest inhibitory effects. This finding revealed that the highest concentrations of phytohormones do not always confer the most resistance to insect herbivores. This finding also agrees with observations made by other studies. For example, Fan et al. [40] treated chrysanthemum plants with five concentrations of MeJA (i.e., at 0.01, 0.05, 0.1, 0.5 and 1 mmol/L) and found that the lowest number of aphids occurred in the 0.05 mmol/L treatment. In another study on cotton treated with 0.1, 1, and 10 mmol/L JA, the 1 mmol/L JA treatment was found to significantly inhibit the relative growth rate and mass gain of *Helicoverpa armigera* [41]. In another study, rice plants treated with 100 or 200 μM ethephon died faster than those treated with water and were found to be less resistant to *Nilaparvata lugens Stål* (BPH) and, moreover, 100 or 200 μM ethephon treatments were also found to increase the body weight of BPH [42]. It is, therefore. possible that the phytohormone concentrations required to activate defense pathways are physiological characteristics specific to different host plant species.

Previous studies have shown that JA and ET synergistically or antagonistically regulate plant defense against herbivorous insects [42]. In this study, we found that treatment with JA followed by ET, ET followed by JA, or simultaneous JA and ET administration (i.e., both sprayed together) could effectively treat faba bean plants, resulting in reduced oviposition and selectivity of thrips. Of these treatments, the JA with ET treatment showed a large and obvious inhibitory effect on the fecundity and feeding selectivity of *F. occidentalis*. This finding suggested that the JA with ET treatment could synergistically promote the defense of faba bean plants against *F. occidentalis*. Similar results have been reported by other studies. For example, JA was found to interact with ET to mediate insect-induced volatile emissions in maize [43], and JA and ET signaling pathways were found to modulate the resistance of *Arabidopsis thaliana* against a leaf-chewing herbivore [44]. These results might be due to the simultaneous activation of the JA and ET pathways, thereby exerting synergistic regulation that enhances plant defense. 

The defense response of faba bean plants treated with exogenous JA and ET was mediated by altered enzyme activity. We found that the LOX and PPO activities in faba bean plants treated with JA and ET significantly increased. We observed higher LOX activity at 24 h, higher AOS activity at 48 h, and higher ACS and TI activities at 72 h compared with the control. Moreover, the relative expression of *LOX1* was significantly upregulated at 24 h, and the expression levels of *AP2/ERF* were significantly upregulated at 48 h. We also found that the relative expression levels of *ACS2* were significantly downregulated at 24 h. Taken together, these results indicated that JA with ET treatment could induce the defense response of faba bean plants to some extent, and different enzymes responded at different times. 

In general, when the activity or expression levels of the enzymes of a host plant defense increase, herbivore resistance also increases [45]. For example, JA application to tomatoes increased the activities of PPO, LOX, and PIs. These increases were associated with an increase in plant chemical defense [6]. In another study, MeJA treatment of Indian mustard induced an increase in PPO activity at 1 d, 3 d, and 5 d, but the authors observed no significant difference at 7 d compared with the control. They also found that the relative expression of *LOX* and *ACO* in the JA synthesis pathway increased significantly at 1, 3, 5, 8, and 24 h [46]. Xin *et al.* [47] found that the relative expression of the LOX synthase gene *CsiLOX1* was significantly higher at 1, 2, 4, and 8 h after treatment, compared with the control, while no significant changes were observed at 0.5, 12, and 24 h after treatment. This study also reported that the relative expression of the ET synthase gene *CsiACS1* was significantly higher at 0.5, 1, and 2 h compared with the control, while no significant changes were observed at 4, 8, 12, and 24 h after treatment. It is worth noting that *AP2/ERF* belongs to the ERF transcription factor family, and its relative expression is regulated by JA and ET [48].

Taken together, our data indicate that changes in defense enzyme activity and JA/ET pathway marker gene expression were induced by a combined JA with ET treatment of faba bean plants. We then compared the two-sex life table of *F. occidentalis* reared by JA with ET-treated and water-treated faba bean plants. The results showed that the treatment of JA with ET prolonged the development of numerous thrips life stages, including the egg, first instar, and immature stages, as well as the adult preoviposition period. Treatment also significantly shortened the *r*, *λ*, and *R*_0_ population parameters and significantly prolonged the *T*. These results indicate that treatment of JA with ET inhibited the growth and development of *F. occidentalis*, at least to a certain extent. It was found in many studies that PPOs and PIs can negatively affect insect digestion, thereby affecting the normal growth and development of insects [49]. For example, Bhonwong et al. [50] reported that PPO overexpression in tomato plants slowed the growth and development of *H. armigera* and *Spodoptera exigua* (Hübner) and reduced their nutritional indices. The intake of PIs is known to impede insect gut digestion, leading to reduced nutrient uptake and amino acid deficiency. This, in turn, leads to developmental delay and reduced fecundity [51]. Moreover, LOX activity might further aggravate the nutritional deficiency of host plants [52]. Therefore, we speculate that the results of this study may be due to JA with ET causing faba bean plants to produce anti-nutritional substances, which could prevent thrips from digesting normally after feeding, thereby negatively affecting normal thrips growth and development.

## 5. Conclusions

In summary, our study demonstrated that an exogenous combined JA with ET treatment had a big influence on thrips feeding preference, fecundity and life parameters. We also found that JA with ET treatment could improve the induced defense of faba bean plants against thrips, by changing enzyme activation and gene expression levels. The results indicated that JA with ET plays a very important role in planned defense against thrips. *F. occidentalis* act as viral vectors, and are known to transmit tospoviruses. Phytohormones also regulate vector virus interactions [53,54]. Thus, exogenous applications of JA and ET might have the effect of inhibiting viral epidemics. Moreover, organic farming nowadays is very popular with less synthetic chemical usage, and, thus, this study has an applied perspective in the current scenario. 

## Figures and Tables

**Figure 1 insects-13-01073-f001:**
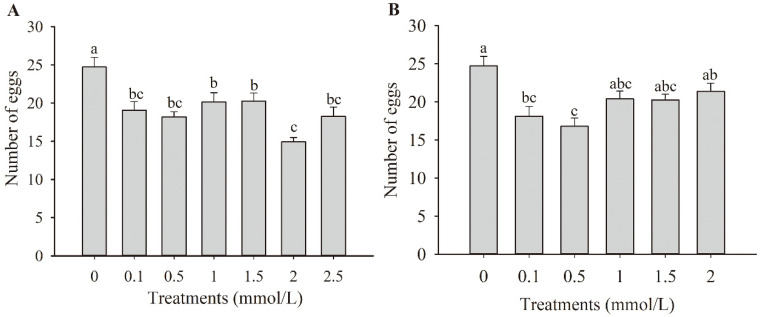
Effects of jasmonic acid (**A**) and ethylene (**B**) treatments on *Frankliniella occidentalis* oviposition in faba bean plants. Data are shown as mean ± standard error (*n* = 15). Different letters above the bars indicate significant differences among different treatments (*p* < 0.05; one-way analysis of variance, Tukey’s test).

**Figure 2 insects-13-01073-f002:**
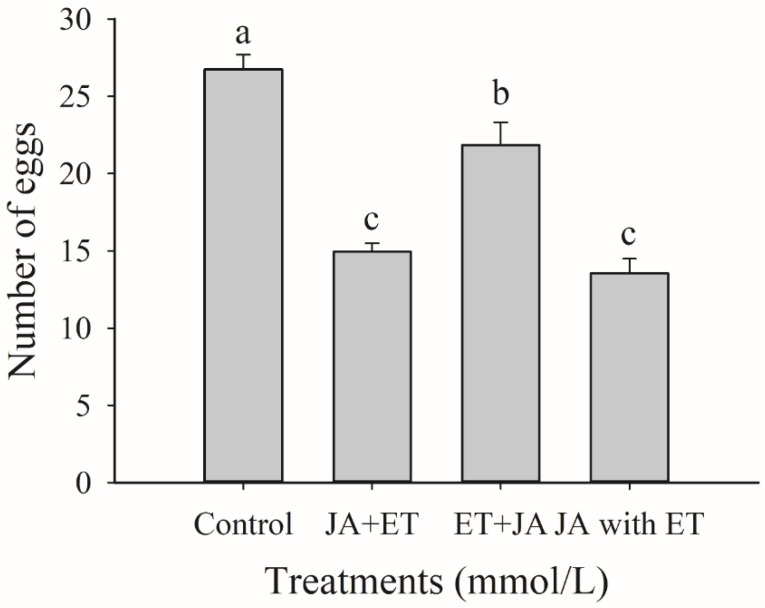
Effect of combined jasmonic acid and ethylene treatments on *Frankliniella occidentalis* oviposition in faba bean plants. Data are presented as mean ± standard error (*n* = 20). Different letters above the bars indicate significant differences among different treatment amounts (*p* < 0.05; one-way analysis of variance, Tukey’s test).

**Figure 3 insects-13-01073-f003:**
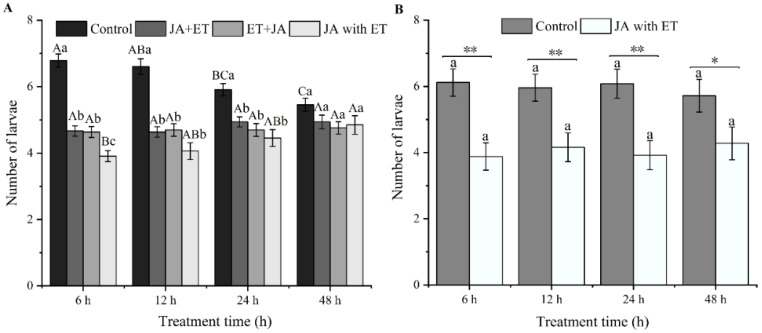
Effects of combined jasmonic acid and ethylene treatments on *Frankliniella occidentalis* spawning in faba bean plants. (**A**): Four different JA/ET treatments. Data are presented as mean ± standard error (*n* = 33). Different capital letters above the bars indicate significantly different mean numbers of thrips among the different time periods of the same treatment, while different small letters indicate significant differences among the different treatments at the specified time period (*p* < 0.05; one-way analysis of variance, Tukey’s test); (**B**) Two treatments (JA with ET and a control). Data are presented as mean ± standard error (*n* = 25). Different small letters above the bars show significant differences at the 0.05 level among different times at the same treatment (Tukey multiple range test); ”*” and “**” indicate significant differences at the 0.05 and 0.01 levels among different treatments for the same time period.

**Figure 4 insects-13-01073-f004:**
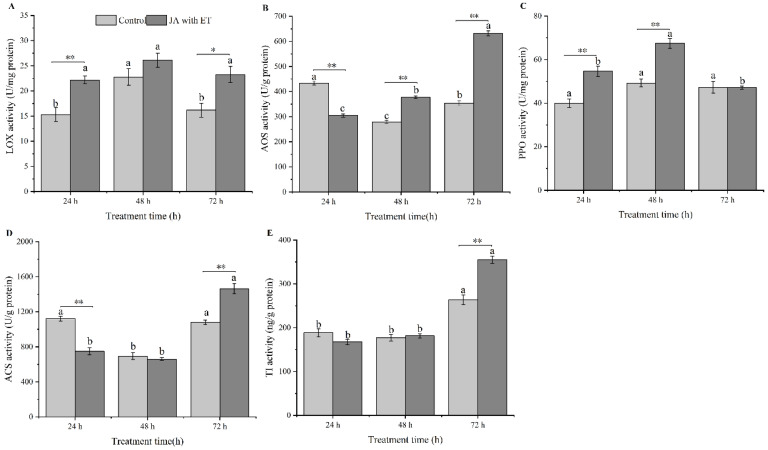
Effect of combined jasmonic acid and ethylene (JA with ET) treatment on the activities of defense enzymes in faba bean plants. (**A**), lipoxygenase (LOX); (**B**), allene oxide synthase (AOS); (**C**), polyphenol oxidase (PPO); (**D**), 1-aminocyclopropane 1-carboxylic acid synthase (ACS); (**E**) trypsin inhibitor (TI). Data are presented as mean ± standard error (*n* = 4). Different small letters above the bars show significant differences at the 0.05 level among different time points of the same treatment (Tukey multiple range test). “*” and “**” indicate significant differences at the 0.05 and 0.01 levels between different treatments at the same time point.

**Figure 5 insects-13-01073-f005:**
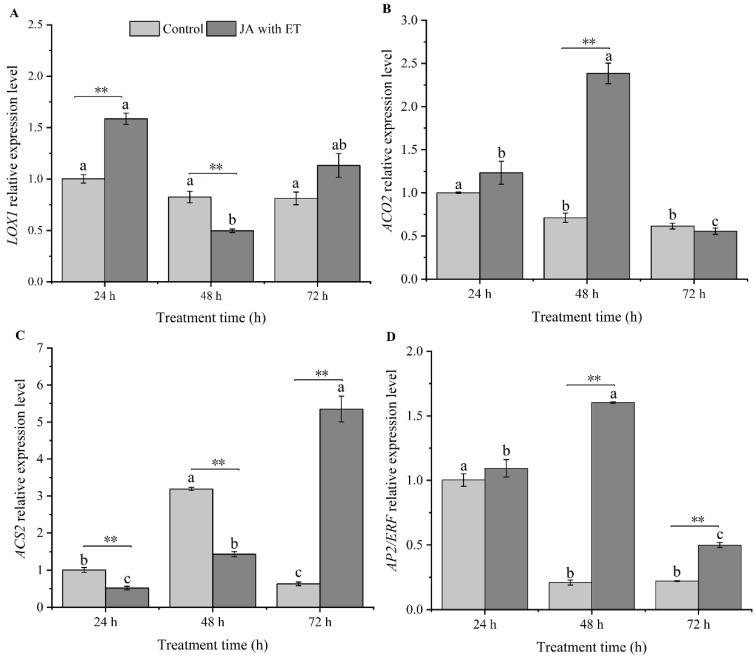
Effect of the JA with ET treatment on the expression of JA/ET pathway marker genes in faba bean plants. (**A**) *LOX1*; (**B**) *ACO2*; (**C**), *ACS2*; (**D**) *AP2/ERF*. Data are presented as mean ± standard error (*n* = 3). Different small letters above the bars show significant differences at the 0.05 level among different time points within the same treatment (Tukey multiple range test). “**” indicate significant differences at the 0.01 levels between different treatments at the same time point.

**Figure 6 insects-13-01073-f006:**
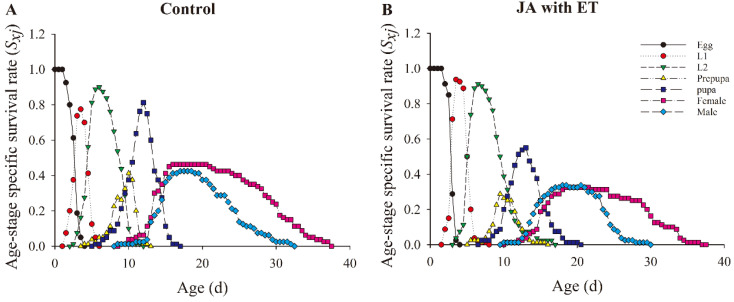
Effect of the JA with ET treatment on the age-stage-specific survival rate (*S_xj_*) of *Frankliniella occidentalis* grown on faba bean plants. (**A**) Control group; (**B**) JA with ET group.

**Figure 7 insects-13-01073-f007:**
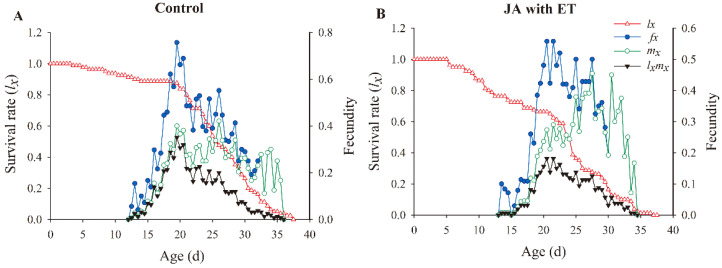
Effect of the JA with ET treatment on the age-specific survival rate (*l_x_*), age-stage-specific fecundity (*f_xj_*), age-specific fecundity (*m_x_*), and age-specific net maternity (*l_x_m_x_*) of *Frankliniella occidentalis* grown on faba bean plants. (**A**) Control group; (**B**) JA with ET group.

**Figure 8 insects-13-01073-f008:**
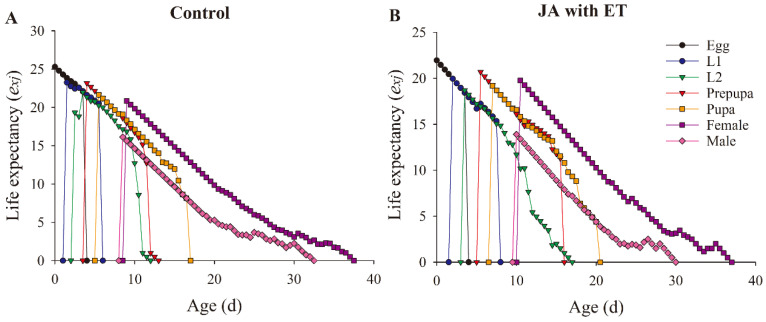
Effect of the JA with ET treatment on the age-stage-specific life expectancy (*e_xj_*) of *Frankliniella occidentalis* grown on faba bean plants. (**A**) Control group; (**B**) JA with ET group.

**Figure 9 insects-13-01073-f009:**
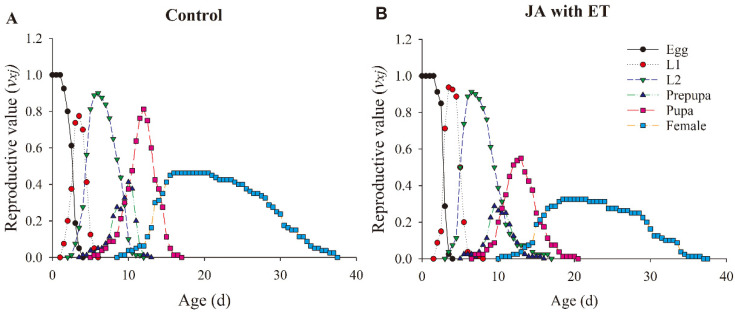
Effect of the JA with ET treatment on the age-stage-specific reproduction value (*v_xj_*) of *Frankliniella occidentalis* grown on faba bean plants. (**A**) Control group; (**B**) JA with ET group.

**Table 1 insects-13-01073-t001:** RT-qPCR genes and specific primers.

Gene	Forward Primers (5′-3′)	Reverse Primers (5′-3′)
*ELF1A*	GTGAAGCCCGGTATGCTTGT	CTTGAGATCCTTGACTGCAACATT
*LOX1*	AGTCCTCAAGTTTCCGCCAC	GGGAGCGAATTAAGCCTGGA
*ACO2*	GACGACAAAGTCAGTGGCCT	TCACTCGGTGCTCTATGCTT
*ACS2*	CAGCCGCAAAATGTCGAGTT	AACCTCTTCGCACTCTCAGC
*AP2/ERF*	CACCGCCGTTTTCTATCTCC	TAACAACGGCAGCGTTTTCA

**Table 2 insects-13-01073-t002:** Effect of the JA with ET treatment of faba bean plants on the development and reproduction of offspring of *Frankliniella occidentalis*.

Index	Control		JA with ET		*p*	
	n	Mean ± SE	n	Mean ± SE		
Egg stage (d)	80	2.79 ± 0.07	80	3.04 ± 0.05	0.0034	**
First instar stage (d)	80	1.73 ± 0.03	78	2.23 ± 0.06	0	**
Second instar stage (d)	73	4.16 ± 0.11	58	4.42 ± 0.18	0.21	ns
Prepupa stage (d)	72	1.38 ± 0.03	57	1.46 ± 0.04	0.11	ns
Pupa stage (d)	71	3.68 ± 0.08	55	3.83 ± 0.10	0.22	ns
Immature	71	13.72 ± 0.19	55	14.91 ± 0.26	0.0002	**
Female adult stage (d)	37	16.42 ± 0.64	26	15.08 ± 0.68	0.15	ns
Male adult stage (d)	34	10.60 ± 0.55	29	9.29 ± 0.36	0.047	*
Per female oviposition	37	12.39 ± 0.57	26	10.08 ± 0.67	0.0098	**
Adult preoviposition period (d)	37	3.05 ± 0.10	26	3.37 ± 0.09	0.025	*
Total preoviposition period (d)	37	16.49 ± 0.28	26	18.56 ± 0.37	0	**
Female longevity (d)	37	29.85 ± 0.70	26	30.27 ± 0.73	0.68	ns
Male longevity (d)	34	24.63 ± 0.59	29	23.95 ± 0.46	0.36	ns
Sex ratio of offspring (%)	929	1.05 ± 0.086	523	1.00 ± 0.96	0.79	ns

Values are shown as mean ± SE. “*” and “**” indicate significant differences at the 0.05 or 0.01 levels between two treatments at the same time point. ns: No significant difference. n: The number of surviving *Frankliniella occidentalis* individuals at a specific time point.

**Table 3 insects-13-01073-t003:** Effect of the JA with ET treatment on the population parameters of offspring of *Frankliniella occidentalis* grown on faba bean plants.

Population Parameter	Control		JA with ET		*p*	
	n	Mean ± SE	n	Mean ± SE		
Intrinsic rate of increase (*r*) (d^1^)	80	0.08 ± 0.01	80	0.05 ± 0.01	0.003	**
Finite rate of increase (*λ*) (d^1^)	80	1.08 ± 0.01	80	1.05 ± 0.01	0.003	**
Net reproductive rate (*R*_0_)	80	5.73 ± 0.74	80	3.30 ± 0.57	0.009	**
Mean generation time (*T*) (d^1^)	80	22.01 ± 0.37	80	23.77 ± 0.42	0.002	**
Gross reproductive rate (*GRR*)	80	10.76 ± 1.03	80	9.20 ± 0.70	0.210	ns

Values are shown as mean ± SE. “**” indicate significant differences at the 0.01 levels between two treatments at the same time point. ns: No significant difference. n: The number of surviving Frankliniella occidentalis individuals at a specific time point.

## Data Availability

The data that support the findings of this study are available from the corresponding author upon reasonable request.

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
