# Peer review of "Combined Jasmonic Acid and Ethylene Treatment Induces Resistance Effect in Faba Bean Plants Against Frankliniella occidentalis (Pergande) (Thysanoptera: Thripidae)"

_insects, 2022, doi:10.3390/insects13111073_

Round 1

Reviewer 1 Report

This seems to me an impressive contribution. Certainly the 'traditional' methods of trying to control this pest, by killing it with various noxious chemicals, are failing. The only objective they achieve is to increase the profits of various companies - and the tax haul for Governments. The lateral thinking involved in searching for ways of altering the behaviour of the pest, and thus inhibiting the development of large populations, is intellectually refreshing - and with considerable economic implications. The objectives are clearly stated, the technical approach and data analysis seem to me to be of an excellent standard. There is a long way to go, but I hope that this paper might stimulate further ideas of how to break out of the pesticide resistance cycle. 

Author Response

Point 1: This seems to me an impressive contribution. Certainly the 'traditional' methods of trying to control this pest, by killing it with various noxious chemicals, are failing. The only objective they achieve is to increase the profits of various companies and the tax haul for Governments. The lateral thinking involved in searching for ways of altering the behaviour of the pest, and thus inhibiting the development of large populations, is intellectually refreshing and with considerable economic implications. The objectives are clearly stated, the technical approach and data analysis seem to me to be of an excellent standard. There is a long way to go, but I hope that this paper might stimulate further ideas of how to break out of the pesticide resistance cycle.

Response 1: We are very grateful for your opinions on this manuscript. We agree that there is a long way to go, and the results of this study will priovide the basis and reference data for controlling of the thrips.

Reviewer 2 Report

In the current manuscript, Jia et al. explored the exogenous application of jasmonic acid and ethylene treatments for inducing host resistance in faba beans against Western flower thrips. The authors have done a tremendous job conducting various experiments and summarizing the outcomes. In my opinion, the manuscript can be considered for publication after making the suggested edits outlined below:

  • Change the word “thrip” to thrips throughout the manuscript
  • Explain how you selected the JA and ET concentrations to start with
  • Address the issue: Error! Reference source not found throughout
  • Update the M & M sections with the number of reps and the number of times each study was conducted
  • Improve the quality of Fig. 3 and 4
  • Discuss if there was any negative impact of treatments on the plant quality and the associated cost comparison in reference to the traditional thrips management strategies.

Author Response

Point 1: Change the word “thrip” to thrips throughout the manuscript.

Response 1: Thanks for your suggestion. We have checked the full text and have changed thrip to thrips.

Point 2: Explain how you selected the JA and ET concentrations to start with.

Response 2: The initial concentrations of JA and ET were based on the concentrations reported by our team's previous results and other reports. For example, the concentration of JA has been reported to be 0.001, 0.01, 1 mmol/L (Li et al., 2017), and the concentration of ET were 0.02, 0.04, 0.06, 0.08 mmol/L (Zhang et al., 2014). Based on the literature, we did a pre-test in which the JA concentration was set at 0.01, 0.05, 0.1, 0.5, 1, 1.5, 2 and 2.5 mmol/L and the ET concentration was set at 0.01, 0.05, 0.1, 0.5, 1, 1.5 and 2 mmol/L. The results showed thatboth JA and ET didin’t have the affect on the oviposition of F. occidentaliswhen concentrations of JA and ET were less than 0.05 mmol/L. So our experimental concentrations started from 0.1 mmol/L.

Point 3: Address the issue: Error! Reference source not found throughout.

Response 3: Thanks for your careful checks. We are sorry for our carelessness. We carefully reviewed the full text and revised the references format. The "Error!...." has been removed.

Point 4: Update the M & M sections with the number of reps and the number of times each study was conducted.

Response 4: Sorry for that we didin’t describe it clealry. We added research repetition and number of times in the materials and methods section. The supplementary content was as follows (or please check the manuscript):

The enzyme activities were calibrated to the protein content of the corresponding sample concentration. There were 4 replicates for each enzyme activity.

All primer sequences were shown in Table 1, and were synthesized by Sangon Biotech Co. td (Shanghai, China). There were three replicates for each gene.

Point 5: Improve the quality of Fig. 3 and 4.

Response 5: Thanks for your suggestion. We have improved the the quality of Fig. 3 and Fig.4.

Point 6: Discuss if there was any negative impact of treatments on the plant quality and the associated cost comparison in reference to the traditional thrips management strategies.

Response 6: Thanks for your suggestion. We have added this part. The supplementary content was as follows (or please check the manuscript):

As important phytohormones, JA and ET play a role in regulating plant adaptation to biotic and abiotic stresses (Yang et al., 2019),and there was no report about negative impact of exogenous JA and ET on the plant quality. In this study, Exogenous JA and ET did not adversely affect the growth and development of faba bean plants. The prices of JA and ET are not higher, so the cost of using exogenous JA and ET is not higher than that of using traditional thrips management strategies. Moreover, pest insects have developed resistance. In order to improve the control effect, the massive chemical pesticides were sprayed, which is more costly and causes pollution to the environment.

Reviewer 3 Report

The study of Jia et al investigates the use of the exogenous application of phytohormones to control Frankliniella occidentalis in faba bean plants. F. occidentalis is an important pest that causes huge crop production loss and needs to be managed efficiently thus, is of importance to the journal readers. It is a preliminary study but an impactful contribution to the field of pest resistance however, there is a scope to improve the current manuscript which is mentioned below.

- The references are not cited in the proper format. The reference number should be in brackets. In some places, it is written "Error!...." should be removed. Please check for this throughout the manuscript.

- line 47: Is it two references 6 and 78 or all references from 6 to 78.

- line 58 to 65: To show the importance of JA in pest-plant interactions. Authors should also indicate not only spraying JA but inhibiting the JA metabolic pathway also causes changes in pest behavior. This highlights the critical role of JA. https://doi.org/10.1007/s10886-020-01157-7

-  line 142: how many hours after JA/Ethylene treatment, thrips were kept?

- Figure 3: line 258 to 261 say that JA with ET treatment was used for later experiments. But, figure 3 again shows data for all the treatments which is confusing. Seems like 3 A and 3B have the same data. Authors should keep one of them if they are the same. 

- Discussion: line 469: Please always refer to the figure numbers that are discussed. Kindly add it throughout the discussion. It helps the readers to understand better.

- To highlight the further importance of their study which helps the readers to understand the broad impact, authors can write a few lines on F. occidentalis acting as viral vectors. They are known to transmit tospoviruses and phytohormones also regulate vector-virus interactions. Thus, exogenous applications can be used to stop viral epidemics. Moreover, organic farming nowadays is very popular with less synthetic chemicals usage thus this study has an applied perspective in the current scenario.  https://doi.org/10.1603/IPM10020; https://doi.org/10.1111/mec.16103; https://doi.org/10.1007/s11356-021-15258-7; https://doi.org/10.1016/j.baae.2022.08.005

Author Response

Point 1: The references are not cited in the proper format. The reference number should be in brackets. In some places, it is written "Error!...." should be removed. Please check for this throughout the manuscript.

Response 1: Thanks for your careful checks. We are sorry for our carelessness. Based on your comments, We carefully reviewed the full text and revised the references format. The "Error!...." has been removed.

Point 2: line 47: Is it two references 6 and 78 or all references from 6 to 78.

Response 2: Thanks again the reviewers for their very careful reviewing the article. We are sorry for our carelessness. The correct format for the reference is [6–8] in line 47.

Point 3: line 58 to 65: To show the importance of JA in pest-plant interactions. Authors should also indicate not only spraying JA but inhibiting the JA metabolic pathway also causes changes in pest behavior. This highlights the critical role of JA. https://doi.org/10.1007/s10886-020-01157-7.  

Response 3: Thanks for your suggestion. We have added this part. The supplementary content was as follows (or please the manuscript):

JA plays a very important role in plant-insect interaction defense against insects. Not only exogenous spraying of JA , but inhibiting the JA metabolic pathway also causes changes in pest behavior. Therefore, it is necessary to seek environmentally friendly control strategies capable of effectively reducing crop yield losses through this approach.

Point 4: line 142: how many hours after JA/Ethylene treatment, thrips were kept?

Response 4: The JA/ET treatment time in the manuscriptwas detailed written in line 120 to 122, and the thrips thrips were kept time in line 149 to 150, which were 6, 12, 14, and 48 h, respectively.

Point 5: l Figure 3: line 258 to 261 say that JA with ET treatment was used for later experiments. But, figure 3 again shows data for all the treatments which is confusing. Seems like 3 A and 3B have the same data. Authors should keep one of them if they are the same.

Response 5: Sorry for that we didin’t describe it clealry. Figures 3A and 3B in our manuscript represented two independent experiments rather than the same data. There were 4 treaments in Figures 3A, and there were only 2 treaments in Figures 3B. The object was to further determine the effect of the JA with ET solution on thrips, the experiment of the selectivity of thrips was assessed just between healthy untreated and JA with ET treated leaves. The supplementary content was as follows(or please see the manuscript):

To further determine the effect of the JA with ET solution on thrips, we assessed another experiment of the selectivity of thrips just between healthy untreated and JA with ET treated leaves in line 152 to 153.

In the experiments that just focusing the effect of JA with ET, the JA with ET treatment was found to significantly affect thrip feeding selectivity compared with the control (Figure 3B) in line 274 to 275.

Point 6: Discussion: line 469: Please always refer to the figure numbers that are discussed. Kindly add it throughout the discussion. It helps the readers to understand better.

Response 6: Based on your comments, We added figure numbers in the discussion. The supplementary content was as follows(or please see the manuscript):

100 or 200μM ethephon treatments were also found to increase the body weight of BPH.

Point 7: To highlight the further importance of their study which helps the readers to  vectors. They are known to transmit tospoviruses and phytohormones also regulate vector-virus interactions. Thus, exogenous applications can be used to stop viral epidemics. Moreover, organic farming nowadays is very popular with less synthetic chemicals usage thus this study has an applied perspective in the current scenario.

Response 7: We appreciate your suggestion. We have supplied in the section of conclusions. The supplementary content was as follows (or please see the manuscript):

This results indicated that JA with ET plays a very important role in plan defense against thrips. However, F. occidentalis acting as viral vectors, they are known to transmit tospoviruses and phytohormones also regulate vector virus interactions. Thus, exogenous applications of JA and ET might have the effect of inhibiting viral epidemics. Moreover, organic farming nowadays is very popular with less synthetic chemicals usage, thus this study has an applied perspective in the current scenario.

Round 2

Reviewer 3 Report

The authors have incorporated changes in the manuscript and improved their work's scientific rigor. 

 In the draft I reviewed, line 47 still shows references 6 - 78. The issues with citations still remain e.g., not in brackets which needs to be taken care of before publication. 

Author Response

Point 1: In the draft I reviewed, line 47 still shows references 6 - 78. The issues with citations still remain e.g., not in brackets which needs to be taken care of before publication.

Response 1: Thanks for your careful checks. Probably due to our computer version problem resulting in insert link error, we have corrected this problem again.